# Soil Microbiome Composition along the Natural Norway Spruce Forest Life Cycle

**Michal Choma** [1,*] , **Pavel Šamonil** [2] **, Eva Kaštovská** [1] **, Jiří Bárta** [1] **, Karolina Tahovská** [1] **, Martin Valtera** [3] **and Hana Šantrůčková** [1]

1    Department of Ecosystem Biology, Faculty of Science, University of South Bohemia, Branišovská 31, 37005 České Budějovice, Czech Republic; ekastovska@prf.jcu.cz (E.K.); barta77@seznam.cz (J.B.); tahovska@centrum.cz (K.T.); hana.santruckova@prf.jcu.cz (H.Š.)
2    Department of Forest Ecology, The Silva Tarouca Research Institute for Landscape and Ornamental Gardening, Lidická 25/27, 60200 Brno, Czech Republic; pavel.samonil@vukoz.cz
3    Department of Geology and Soil Science, Faculty of Forestry and Wood Technology, Mendel University in Brno, Zemědělská 3, 61300 Brno, Czech Republic; martin.valtera@mendelu.cz
*    Correspondence: michal.choma@prf.jcu.cz

**Abstract:** Stand-replacing disturbances are a key element of the Norway spruce (*Picea abies*) forest life cycle. While the effect of a natural disturbance regime on forest physiognomy, spatial structure and pedocomplexity was well described in the literature, its impact on the microbiome, a crucial soil component that mediates nutrient cycling and stand productivity, remains largely unknown. For this purpose, we conducted research on a chronosequence of sites representing the post-disturbance development of a primeval Norway spruce forest in the Calimani Mts., Romania. The sites were selected along a gradient of duration from 16 to 160 years that ranges from ecosystem regeneration phases of recently disturbed open gaps to old-growth forest stands. Based on DNA amplicon sequencing, we followed bacterial and fungal community composition separately in organic, upper mineral and spodic horizons of present Podzol soils. We observed that the canopy opening and subsequent expansion of the grass-dominated understorey increased soil N availability and soil pH, which was reflected in enlarged bacterial abundance and diversity, namely due to the contribution of copiotrophic bacteria that prefer nutrient-richer conditions. The fungal community composition was affected by the disturbance as well but, contrary to our expectations, with no obvious effect on the relative abundance of ectomycorrhizal fungi. Once the mature stand was re-established, the N availability was reduced, the pH gradually decreased and the original old-growth forest microbial community dominated by acidotolerant oligotrophs recovered. The effect of the disturbance and forest regeneration was most evident in organic horizons, while the manifestation of these events was weaker and delayed in deeper soil horizons.

**Keywords:** soil microbiome; *Picea abies*; forest disturbance; forest life cycle; Podzol; soil genesis





## 1. Introduction

In many European mountains, the tree line is formed by forest stands dominated by Norway spruce (*Picea abies*, (L) Karst.). In the prehistorically inhabited central Europe, only few remnants of primeval spruce mountain forests have been preserved till the current time [1]. The life cycle dynamics of these forest ecosystems is driven by a specific disturbance regime, which is in some aspects more similar to the dynamics of northern boreal forests than of the temperate mixed and deciduous forests in lower elevations. In old-growth, spruce-dominated mountain forests, stand-replacing disturbances are characteristically caused by abiotic (e.g., windstorm) and biotic (typically an outbreak of spruce bark beetles: *Ips typographus* L., *Ips duplicatus* Sahlberg) factors [2,3]. Although fire events cannot be excluded in general, these were common, particularly in early and middle Holocene [4,5]. Occasional uprooting and breaking of single to several trees or the

whole-stand disturbance during storms usually followed by an outbreak of bark beetles can result in canopy openings on scales from small gaps in the size of fractions of a hectare to large areas spanning over thousands of hectares [6–9]. The disturbance regime of mountain spruce forests shapes their physiognomy, spatial structure, pedocomplexity and even the landform form [10] by enabling re-occurrence of early phases of forest succession and their further development towards the re-establishment of the old-growth forest [3].

The tree-layer breakdown is followed by decades of gradual forest re-establishment. In the regeneration phase, the original shady, compact forest vegetation is opened, brightened and warmed. Thick forest floor horizons accumulated for decades are gradually mineralized, and formerly fixed nutrients are released [11,12]. Frequently, in the period of reduced competition in the tree layer and released resources (nutrients, light and water), the grass vegetation (*Calamagrostis villosa*, *Nardus stricta*) and occasionally ferns (*Athyrium distentifolium*) or blueberry (*Vaccinium myrtillus*) develop massively. The character of forest vegetation changes into meadow-like for decades [13]. After that, the growing young forest effectively fixes the majority of released nutrients and gradually shades.

Temporary forest-for-grass vegetation substitution may change the trajectory of soil evolution. A humid cold climate, acidic bedrock, gentle-to-steep slopes and acidifying spruce vegetation commonly lead to the formation of Podzols. This soil unit is typical formed by an evolution of thick upper organic horizons (L-litter, F-fermented and H-humification) [12], a strongly acidic upper mineral A horizon and sharply separated eluvial E versus illuvial spodic Bhs horizons, which largely differ in their chemical properties [14]. The individual soil horizons influence each other, but at the same time, they have different dynamics and thus different memory [15]. While the material in the upper horizons is completely replaced in the order of years to the first decades, the pedogenetical memory of the lower Bs horizon can last for centuries or millennia. Hypothetically, therefore, we can assume that a change in the vegetation of a mountain forest will cause a kind of polygenetic wave, which will gradually spread to deeper horizons. Potentially, a regressive genesis of Podzols might occur. This process was described as a result of anthropogenic intervention [16] but never as a component of natural cycle of a terrestrial ecosystem. Šamonil et al. [17] found in the evolution of Podzols little support for regressive development without their mechanical disturbance (e.g., by tree uprooting). Instead of a real regression of soil development, it could only be a colour hiding (or overlaying) of the white eluvial E horizon caused by leaching organic substances produced in the upper soil layer due to a vegetation composition shift, without significant changes in subsoil chemistry.

The vegetation structure and composition affect microclimate, soil properties [18] and the species-specific plant microbe selection, factors that mutually shape the composition of the litter and rhizosphere microbiome [19–21]. Mature Norway spruce stands create specific soil conditions that select for a characteristic microbial community. Particularly, the input of low-quality spruce litter leads to low-pH and nutrient-poor conditions in organic and upper mineral horizons that select for acidotolerant and oligotrophic microbes [22–24]. The fungal community is characteristically dominated by ectomycorrhizal oligotrophic fungi, directly fed by trees and capable of decomposing recalcitrant substrates poor in nutrients, which prevail over saprotrophs [25–27]. However, the stand-replacing disturbances and shift in the vegetation composition deflect the system from this state and change the factors that govern the soil microbiome composition. The tree death enables the expansion of grasses and ferns that produce faster decomposable, nutrient-richer litter and root exudates, which stimulate the decomposition of previously accumulated spruce litter, enhance soil nutrient availability and might increase soil pH [28–30]. The soil pH and nutrient status, represented especially by the nitrogen (N) content and the carbon:nitrogen ratio (C:N), are regarded as the most influential factors driving the microbial community composition in forests [20,31–33]. Enhanced N availability following stand-replacing disturbance strengthens the competitive pressure on oligotrophs and enables the expansion of nitrophilic ectomycorrhizal fungi, saprotrophic fungi and bacteria [25,26] and later also

other microbes dependent on high N availability (i.e., nitrifiers). The increased pH releases the acidity-driven physiological stress, which is a crucial determinant of microbial abundance and diversity, especially in the case of bacteria [22,34]. Besides soil chemistry, the microbial diversity further depends on the presence of special niches such as viable tree roots (ectomycorrhizal fungi and other ectomycorrhizosphere fungi and bacteria), spruce needle litter (litter saprobes) or deadwood (white rot fungi) [35,36]. The presence of these microenvironments changes with the progression within the forest life cycle [37]. After disturbance, vegetation regeneration is parallelly followed by a succession of the soil microbial community until the re-establishment of the mature forest. The microbiome succession further continues in compact, shady forests, but its dynamics is weaker compared to the forest regeneration phases [20,38,39].

Regarding the coniferous forest life cycle, particular emphasis has been put on fungi, especially ectomycorrhizal fungi. Numerous studies have reported the retreat of ectomycorrhizal fungi and the prevalence of ruderal species after tree removal and subsequent recovery of ectomycorrhizal dominance over the soil microbial community and a shift back towards stress-tolerant competitive species characteristic of old-growth forests, e.g., [38,40–42]. Despite the high value of such studies, these observations come from plantations, where the forest dynamics is driven by clear-cutting, artificial tree replanting and further silvicultural treatments, which puts the extent of the comparability with the natural regime in question. Less frequent reports of the impact of natural disturbances on the fungal community reveal an analogously negative reaction of the ectomycorrhizal community [37,43,44]. However, the aspect of further progress of these communities with forest regeneration is under-represented in the literature. Moreover, soil bacteria are widely overshadowed by fungi in studies of such topics despite their non-neglectable role in forest ecosystems [22,45]. This study, therefore, aims to shed more light on the development of both bacterial and fungal communities in the context of changes in vegetation and soil properties during the ecosystem progression in the natural Norway spruce forest life cycle.

Additionally, specific horizons within the Podzol soil profile host different microbial communities [46,47], which are expected to be exposed to changes in ecosystem properties to a different extent. In detail, topsoil organic and upper mineral horizons are under the direct influence of litterfall and are densely rooted and explored by mycorrhizal and saprotrophic fungi; thus their microbiome should be the most heavily affected by the vegetation cover change. Only a small fraction of organic matter from the decomposing litter leaches to mineral spodic horizons where the roots are sparsely distributed, which implies a lower dependence of the microbial community on the vegetation status in this part of the soil profile. For that reason, we attempt to trace the soil and soil microbiome development separately in individual soil horizon layers.

The main goal of our study is to reveal compositional changes in soil microbial communities at different phases of the primeval mountain spruce forest life cycle driven by stand-replacing disturbances. Moreover, we aim to identify factors that govern the microbiome shifts in different soil horizons across the sites under study. For this purpose, we carefully selected a chronosequence of six representative sites with increasing time passed since the last stand-replacing disturbance (16, 36, 100, 110, 130 and 160 years). In the chronosequence, the 16- and 36-year sites are still forest gaps with grassy vegetation cover, and the other four sites are closed-canopy mature forest stands. At each site, samples were taken using a soil probe separately from the respective surface organic, upper mineral and spodic horizons. Each soil sample was characterized for microbiome composition by soil DNA amplicon sequencing and for chemical properties in an attempt to test the following hypotheses:

(1)  Microbial communities will differ along the gradient of time since the last disturbance. The communities of regenerating sites will be less contributed by ectomycorrhizal fungi due to a loss of their tree partners but more represented by copiotrophic saprobes and other nitrophilic groups compared to the old-growth forests, where these microbial groups will be restricted by low pH and low N availability. We expect that the

changes will be the most evident in the youngest phases of forest regeneration and the dynamics will be flattened with the consolidation of the closed forest.

(2) Fungal and bacterial communities will be shaped by different factors. The time since the last disturbance (i.e., the advancement of succession towards the old-growth forest) will be the key factor for the fungal community, especially for the ectomycorrhizal fungi dependent on their host trees, while bacteria will be largely determined by the soil pH and the C:N ratio.

(3) The temporary change in the character of vegetation due to disturbance will be manifested in the soil by a polygenetic wave, i.e., changes in the soil most strongly and immediately affect the upper horizons; towards the depth the effect will be delayed and flattened. Accordingly, the effect of time since the last disturbance on the microbiome composition will be the most evident in organic horizons and will decrease with soil depth.

## 2. Materials and Methods

### 2.1. Study Sites and Soil Sampling

The research took place in the volcanic Calimani Mts. in the Inner Eastern Carpathians, northern Romania (47°6.115′ N 25°7.129′ E), in July 2015. The studied well-preserved primeval forest ecosystems belong to the Calimani National Park. The average annual temperature varies between 2.4 and 4.0 °C, and the average annual precipitation varies between 970 and 1150 mm and increases with the latitude [48]. Forest ecosystems in an altitudinal range between ca. 1200 and 1700 m a.s.l. are characteristically dominated by *Picea abies* with a less admixture of *Sorbus aucuparia* and occasionally *Larix decidua*, *Pinus cembra*, *Betula pendula* and *Acer pseudoplatanus*. The forest understorey consists particularly of *Calamagrostis villosa*, *Vaccinium myrtillus*, *Athyrium distentifolium*, *Luzula sylvatica* and *Avenella flexuosa*. In areas occupied by mature, compact spruce forest stands, litter free of the understorey appears. Plant nomenclature follows the taxonomy by Kubát et al. [49].

Using dendrochronological techniques, Svoboda et al. [50] uncovered the detailed disturbance history of primeval *Picea abies* forests in the Calimani National Park core zone. In 5 separate primeval forest remnants, the authors established and assessed in total 82 circular research plots, each of an area of 500 or 1000 m$^2$. Based on processing of the extracted thousands of core series, the authors concluded that the mountain forest dynamics was driven by infrequent stand-replacing disturbances, and they even determined the age since the last strong event per research plot. Whereas the majority of plots were strongly disturbed in the period 1900–1930, a significant part was affected formerly between 1840 and 1870 and a few plots in some other time. Simultaneously, Valtera et al. [51] described soil diversity within the largest primeval forest remnant (ca. 40 ha). They revealed a surprisingly high soil diversity and soil spatial variability. In 360 shallow soil profiles (9 per plot), Entic Podzols predominated, followed by Albic Podzols, Dystric Cambisols and Leptosols (international soil taxonomy according to Michéli et al. [52]). Ranges of spatial correlation of soil properties were comparatively short on a scale of meters or maximally tens of meters, and tree individuals were recognized to be an important source of variability in the forest-floor horizons. Valtera et al. [53] deepened the issue of spatial soil variability in the Calimani Mts. and partitioned it between the effects of site, topography, forest stand and other sources. Finally, Valtera and Šamonil [54] evaluated the course of pedogenetical pathways and carbon stocks in these unique primeval forests. Based on the above-mentioned former research activities, it was possible to precisely select research sites for our current research. Selected sites well represent the phases of the mountain forest life cycle as well as the local variability in soils.

We selected a chronosequence of six representative sites with increasing time passed since the last stand-replacing disturbance (16, 36, 100, 110, 130 and 160 years; Figure 1, Table 1). Within the selected site, we found a position representative from the viewpoint of soil variability [51]. Only recent gaps including the two youngest sites could not be part of the previous research for methodological reasons. Basic dendrochronological and

pedological research was carried out additionally on these sites in 2015 before sampling. While the 16- and 36-year sites after the disturbance represented a young forest of opened canopy with an understorey formed by grasses (mostly *Calamagrostis villosa*, but also with occasional contribution of *Nardus stricta*, *Avenella flexuosa* and *Athyrium distentifolium*), sites disturbed in the older past already represented spruce forests of various degrees of canopy compaction.

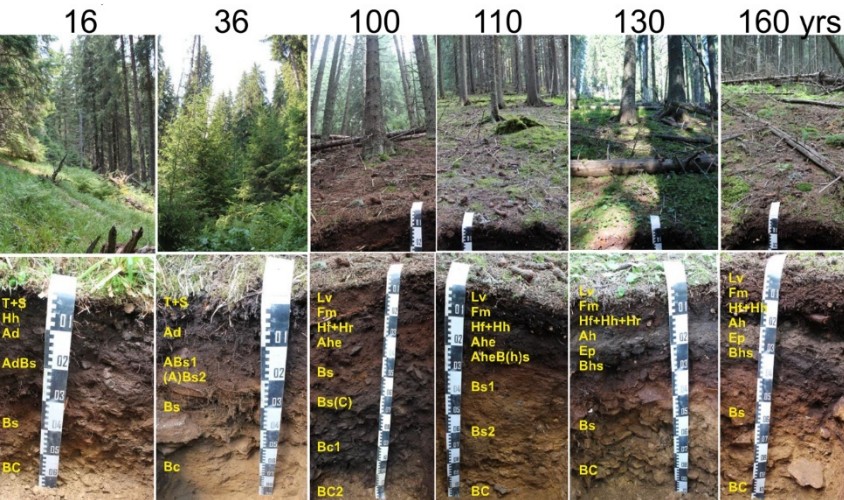

**Figure 1.** Study sites and representative soil profiles; the soil profiles were classified according to the international World Reference Base for Soil Resources (WRB) [52]; yellow letters denote the particular soil horizons [55] and their position the horizontal centre of the particular sampling interval. Note that the perspectives of the original photographs have been modified. Photos by P.Š.

**Table 1.** Description of site vegetation (time since the last stand-replacing disturbance, canopy closure) and soil profile (soil unit, diagnostic horizons and their grouping for the purpose of this study).

|  | 16 Years | 36 Years | 100 Years | 110 Years | 130 Years | 160 Years |
|---|---|---|---|---|---|---|
| Time since the last stand-replacing disturbance [a] (years) | 16 | 36 | 100 | 110 | 130 | 160 |
| Canopy closure (%) | 0 | 20 | 95 | 94 | 89 | 83 |
| Soil unit | Entic Podzol | Entic Podzol | Entic Skeletic Andic Podzol | Entic Podzol | Albic Skeletic Podzol | Albic Podzol |
| Upper organic (O) horizons | T | T | L + F | L + F | L + F | L + F |
|  | H |  | H | H | H | H |
| Upper mineral (A) horizons | A | A | A | A | A | A |
|  | AB | AB |  | AeB | Ep | Ep |
| Spodic (B) horizons | AB | (A)Bs | Bs | Bs | Bhs | Bhs |
|  | Bs | (A)Bs | Bs | Bs | Bs | Bs |

[a]—according to growth release; 36-year plot according to minimal age.

Standard soil profiles excavated by hand down to substratum horizons were described in terms of soil morphology [56] and classified according to the international World Reference Base for Soil Resources (WRB) [52]. Upper organic horizons and humus forms were classified according to Klinka et al. [57] and Jabiol et al. [12] (Figure 1, Table 1). We combined sampling by a soil corer (diameter 6 cm) with the sampling in excavated profiles. We sampled separately specific horizons (including upper organic horizons) and depths; horizons were never mixed during the sampling [54]. We differentiated three types of organic (O) horizons based on their character: L + F (combined L-litter and F-fermented layers of mature forest stands 100–160 years), T (grass-derived litter in gaps of 16 and 36 years) and H (H-humification layer, present in all sites except 36 years) and sampled separately each of these types that were present in the study sites. We sampled 3 replications per site and depth in the soil profile. Extracted samples were kept cool until laboratory analysis.

Subsamples for DNA extraction were air-dried within one day since sampling and frozen at $-80\ ^{\circ}C$ upon arrival to laboratory.

### 2.2. Soil Chemistry Analyses

Soil samples were analyzed in terms of soil chemistry according to Zbíral [58,59] and Zbíral et al. [60]. We evaluated the exchange soil reaction (pH) using 0.2 M KCl. Characteristics of the sorption complex, i.e., concentrations of $Ca^{2+}$, $Mg^{2+}$, $K^+$ and $Na^+$, cation-exchange capacity (CEC, effective) and exchangeable acidity (Al + H)—all using the $BaCl_2$-compulsive exchange procedure and native pH according to Gillman and Sumpter [61]. The total organic carbon content ($C_{tot}$) was measured using the spectrophotometric approach after oxidation by $H_2SO_4$ + $K_2Cr_2O_7$ and the total nitrogen content ($N_{tot}$) according to Kjeldahl [62]. Dissolved organic carbon (DOC) and dissolved nitrogen (DN), which represent microbially available forms, were extracted with distilled water (soil:water 1:10, *w/v*, shaking for 60 min/150 opm, centrifugation at 4000 rpm/10 min, filtered through acid-washed a 0.45 µm glass-fibre filter), and C and N concentrations in water extracts were measured using LiquiTOC II (Elementar, Germany). $C_{tot}$:$N_{tot}$ and DOC:DN ratios were calculated on a molar basis. The amount of phosphorus in soils was studied according to Bray [63,64].

### 2.3. Soil Microbial Communities

Total soil DNA was extracted according to the protocol by Urich et al. [65], followed by a purification using the CleanAll DNA/RNA Clean-up and Concentration Micro Kit (Norgen Biotek Corp., Thorold, ON, Canada). The concentration of obtained DNA was quantified using SybrGreen [66]. Bacterial and fungal marker genes were quantified, as described previously [26]. Briefly, primer pairs 341F+534R and nu-SSU-0817-50+nu-SSU1196-30 were used to target bacterial 16S rRNA and fungal 18S rRNA genes, respectively [67,68]. The marker gene abundance was quantified on The StepOne™ Real-Time PCR System (Life Technologies, Waltham, MA, USA) using a serial dilution of the known amount of a purified PCR product obtained from genomic *Escherichia coli* and *Aspergillus niger* DNA as standards in the case of bacteria and fungi, respectively. The fungi-to-bacteria (F:B) ratio was calculated as the sum of 18S copies divided by 16S copies.

The microbial communities were characterized by barcoded amplicon sequencing using the Illumina MiSeq platform (ARGONE Lab, Illinois, IL, USA). The analysis targeted the bacterial V4 region (primers 515F/806R) and the fungal ITS1 region (primers ITS1F/ ITS2) [69,70]. A detailed description of the library preparation and sequencing was published previously [26].

Bacterial paired-end reads were merged using USEARCH 10 [71], while only forward reads were analyzed in the case of fungi. Raw reads were quality-filtered (min. quality mean 25, no ambiguous bases; bases with quality of <25 trimmed at the right end of fungal reads prior to filtering) using PRINSEQ-lite 0.20.4 [72]. ITSx 1.0.11 was used to extract the ITS1 region from fungal reads [73], and the obtained ITS1 sequences were length-filtered to a minimal length of 150 bp. Bacterial sequences were filtered and trimmed to a length of 250 bp. OTUs were clustered at 100% and 98.5% similarity for bacteria and fungi, respectively (singletons discarded), with the use of USEARCH 10 [71]. Taxonomy was assigned by the BLAST algorithm [74] through the parallel_assign_blast.py script of the QIIME v 1.9.1 pipeline [75] against Silva 138 [76] and UNITE 7.2 [77] databases for bacteria and fungi, respectively. OTU tables were further processed with the phyloseq package 1.22.3 [78] for R [79]. A minimal count of 2000 reads per sample is regarded to sufficiently cover the main pattern in the microbial community [69]; thus 18 and 2 samples with a lower sampling depth were omitted from the fungal and bacterial datasets, respectively.

The lifestyle of fungal OTUs with assigned genus or species identity was searched in FUNGuild database v1 [80] and in the available literature, e.g., [32,81,82]. PICRUSt2 was used to predict the relative abundance of prokaryotic OTUs potentially participating in metabolic N transformations [83]. Relative abundances of OTUs that were predicted

to possess KEGG orthologs (as defined by Kyoto Encyclopedia of Genes and Genomes) for genes participating in key transformations within the N cycle: first step of nitrification (K10944, K10945 and K10946—genes amoA, amoB and amoC, respectively), first step of nitrate respiration (K00370, K00371, K00374, K02567 and K02568—genes narG, narH, narI, napA and napB, respectively) and N fixation (K02586, K02591, K02588, K02896, K22897, K22898 and K22899—genes nifD, nifK, nifH, anfG, vnfD, vnfK, vnfG and vnfH, respectively) were summed. Sequence data are stored in the European Nucleotide Archive (https://www.ebi.ac.uk/ena) under the study accession number PRJEB43334.

### 2.4. Statistical Analysis

All steps of the statistical evaluation were performed in R 3.6.1 [79]. The results were considered significant at $p < 0.05$. Sequences were rarefied to a minimal sequencing depth in each dataset (4781 and 2251 for bacteria and fungi, respectively) for alpha diversity analyses, but all other calculations (beta diversity based on the Bray–Curtis sample dissimilarity, taxa relative abundances, etc.) were performed on nonrarefied datasets [84]. Alpha diversity measures: OTU richness and the Shannon index (Magurran, 2004) were calculated at the OTU level.

The samples from the respective diagnostic horizons and depths were grouped into three soil layers: organic (O), upper mineral (A) and spodic (B) horizons (Table 1). These groups were further treated separately. The effect of the site (representing the time from the last disturbance) on the soil chemistry and the relative abundance of microbial groups in the respective horizons and the effect of the organic horizon type (T, L + F and H) in O horizons were tested using one-way ANOVA. After square-root transformation, the relative abundance of fungal genera and/or orders and that of bacterial classes and the relative abundance of microorganisms potentially capable of nitrification, nitrate respiration and N fixation were subjected to analysis. Post hoc comparison of sites was done using estimated marginal means—emmeans v.1.4.1 [85]. The site differences between the overall bacterial and fungal communities were examined by PERMANOVA analysis (9999 permutations) of Bray–Curtis dissimilarities at the OTU level using the package vegan v.2.5-4 [86]. Prior to analysis, the homogeneity of multivariate dispersions was checked. A pairwise PERMANOVA comparison was performed using RVAideMemoire v.0.9-72 [87]. The soil chemical variables best explaining the variability in microbial communities, across the sites, were identified by forward selection in distance-based redundancy analysis (db-RDA). Non-collinearity of selected variables was controlled by exploring the variance inflation factors. The selected variables were then used to perform canonical correlation analysis (CCA) on matrices of relative abundances of bacterial and fungal OTUs in the respective horizons.

## 3. Results

### 3.1. Soil Morphology and Classification

Diagnostic spodic horizons were present, and podzolization was the main soil-forming process in all research sites. Some profiles also showed andic properties (e.g., low bulk density). The gradient of time since the high-severity disturbance corresponded surprisingly well with the gradient of soil formation. Towards sites disturbed in the deeper past, the intensity of podzolization increased, manifested by the presence and thickness of the eluvial E and the illuvial spodic Bhs horizons. While soils on recently disturbed sites had a thick upper mineral A horizon covered by grasses and were classified as Entic Podzols, soils at the opposite end of the gradient showed compact and thick E and Bhs horizons, were covered by a thick litter of spruce needles and were classified as Albic Podzols (Figure 1, Table 1). Other soil profile properties complemented this gradient well—site 110 years expressed soil at the transition of Entic and Albic Podzols, while previously disturbed sites showed a clear E horizon. Base chemical soil properties corresponded well with soil morphology. The soil pH and cation-exchange capacity decreased in thickening E horizon, as described previously [54].

### 3.2. Soil Chemistry

Generally, all the soils were acidic (exchangeable pH 2.5–4.1). Soil pH increased with soil depth in the individual sites and decreased with time since the last disturbance (referred to as time thereafter) in A and B horizons across sites (Table 2). The pH significantly correlated with time in A horizons (r = −0.74, *p* = 0.009), but the differences between sites were not significant. The correlation of the pH with time in B horizons was marginally insignificant (r = −0.55, *p* = 0.07). Dissolved N (DN) concentration was higher in O and A horizons in gaps compared to mature forests (site differences ANOVA F = 4.67, *p* = 0.003 and F = 2.81, *p* = 0.036 for O and A horizons, respectively; Table 2). Soil dissolved organic C (DOC) to DN ratio (DOC:DN) increased along the chronosequence in all horizons (F = 8.23, *p* < 0.001; F = 12.6, *p* < 0.001 and F = 22.2, *p* < 0.001 for O, A and B horizons, respectively; Table 2). In O horizons, the increase was gradual from the 16-year site with the ratio of 7.8 to 21 in 130 and 160-year sites, respectively. In A and B horizons, the DOC:DN in 16y was higher than in 36y site, where the ratio was lowest across all studied sites and it further gradually increased with the time since disturbance (Table 2).

**Table 2.** Soil-exchangeable pH (pH), total carbon ($C_{tot}$) and total nitrogen ($N_{tot}$) concentration (mg g$^{-1}$), their molar ratio ($C_{tot}$:$N_{tot}$), dissolved organic carbon (DOC) and dissolved nitrogen (DN) concentration (µg g$^{-1}$), their molar ratio (DOC:DN) and cation-exchange capacity (CEC; µmol cheq g$^{-1}$) in the respective sites and in their O, A and B horizons. Values represent the mean. Small letters denote differences significant at *p* < 0.05.

| | | Horizons | 16 Years | 36 Years | 100 Years | 110 Years | 130 Years | 160 Years |
|---|---|---|---|---|---|---|---|---|
| pH | | O | 3.0 | 3.2 | 3.0 | 3.0 | 2.8 | 2.6 |
| | | A | 3.1 | 3.4 | 3.0 | 3.0 | 2.6 | 2.5 |
| | | B | 3.8 ab | 4.1 a | 4.0 a | 3.7 ab | 4.1 a | 3.4 b |
| $C_{tot}$ | mg g$^{-1}$ | O | 34.9 b | 39.0 ab | 44.2 a | 44.7 a | 43.8 a | 39.7 ab |
| | | A | 8.5 | 10.7 | 23.8 | 14.1 | 11.5 | 11.6 |
| | | B | 7.0 bc | 6.9 bc | 17.8 a | 8.9 b | 6.3 bc | 4.0 c |
| $N_{tot}$ | mg g$^{-1}$ | O | 1.86 | 2.02 | 1.63 | 1.64 | 1.73 | 1.63 |
| | | A | 0.52 | 0.59 | 1.24 | 0.69 | 0.65 | 0.62 |
| | | B | 0.33 b | 0.27 b | 0.71 a | 0.33 b | 0.38 b | 0.21 b |
| $C_{tot}$:$N_{tot}$ | mol:mol | O | 21.8 c | 22.8 bc | 31.6 a | 32.1 a | 29.6 a | 28.5 ab |
| | | A | 19.5 | 21.8 | 22.4 | 23.9 | 20.4 | 25.3 |
| | | B | 25.6 b | 30.3 ab | 32.0 a | 30.8 ab | 28.2 ab | 27.3 b |
| DOC | µg g$^{-1}$ | O | 2757 | 3277 | 3365 | 3737 | 4262 | 3259 |
| | | A | 603 | 543 | 1454 | 1011 | 1110 | 1147 |
| | | B | 319 | 214 | 294 | 282 | 301 | 328 |
| DN | µg g$^{-1}$ | O | 372 ab | 454 a | 198 bc | 196 bc | 205 abc | 159 c |
| | | A | 42.2 a | 89.2 a | 99.4 a | 48.1 a | 51.6 a | 51.4 a |
| | | B | 18.5 | 19.3 | 19.7 | 16.6 | 12.6 | 11.8 |
| DOC:DN | mol:mol | O | 7.8 c | 9.4 bc | 17.4 ab | 20.4 a | 21.5 a | 21.2 a |
| | | A | 16.8 a | 6.7 b | 16.3 a | 21.1 a | 21.8 a | 23.4 a |
| | | B | 17.4 cd | 12.5 d | 15.6 cd | 17.5 c | 24.1 b | 28.2 a |
| CEC | µmol cheq g$^{-1}$ | O | 312 | 216 | 303 | 308 | 260 | 298 |
| | | A | 324 ab | 259 c | 363 a | 297 bc | 204 d | 129 e |
| | | B | 212 | 140 | 217 | 184 | 135 | 166 |

### 3.3. Soil Microbial Communities

The curated datasets comprised 794,964 and 1,777,403 fungal and bacterial reads, respectively. The per sample values ranged between 2251 and 46,243 and between 4781 and 24,849 sequences for fungi and bacteria, respectively. Fungal amplicons clustered into 2337 OTUs, while 7457 bacterial OTUs were discovered. Fungal diversity indices within respective horizons were comparable among sites, but in the case of the bacterial community, both diversity indices were comparable in O horizons but significantly decreased with

stand age within A and B horizons (OTU richness: F = 4.49, $p$ = 0.004 and F = 6.88, $p$ < 0.001; Shannon index F = 4.83, $p$ = 0.003 and F = 5.20, $p$ = 0.001 for A and B horizons, respectively; Table 3). Fungal marker gene (18S rRNA gene) abundances in respective horizons were similar among sites with an exception of a lower value in the 160-year site than in the other sites in A horizons (F = 5.40, $p$ = 0.001; Table 3). On the contrary, the bacterial marker (16S rRNA gene) was more abundant in 16- and 36-year sites than the old-growth forest sites in all horizons (F = 9.51, $p$ < 0.001; F = 11.7, $p$ < 0.001; F = 3.99, $p$ = 0.007 for O, A and B horizons, respectively; Table 3).

**Table 3.** Site differences for fungal and bacterial diversity measures and marker gene abundances. Values represent the mean with standard deviation in parentheses. Small letters denote differences significant at $p$ < 0.05.

| | Horizon | 16 Years | 36 Years | 100 Years | 110 Years | 130 Years | 160 Years | ANOVA Site F | ANOVA Site p |
|---|---|---|---|---|---|---|---|---|---|
| Fungi OTU richness OTUs | O | 127 (41) | 143 (11) | 84 (37) | 131 (28) | 91 (31) | 127 (12) | n.s. | |
| | A | 95 (18) | 96 (30) | 90 (6) | 86 (18) | 77 (17) | 64 (22) | n.s. | |
| | B | 78 (19) | 72 (19) | 75 (17) | 102 (12) | 81 (28) | 91 (24) | n.s. | |
| Fungi Shannon | O | 2.86 (0.45) | 3.22 (0.10) | 2.32 (0.86) | 3.15 (0.34) | 2.36 (1.04) | 3.00 (0.21) | n.s. | |
| | A | 2.16 (0.11) | 1.90 (0.81) | 2.48 (0.26) | 2.42 (0.34) | 2.53 (0.41) | 2.44 (0.76) | n.s. | |
| | B | 2.20 (0.38) | 2.45 (0.29) | 2.45 (0.51) | 2.58 (0.35) | 2.38 (1.25) | 2.57 (0.56) | n.s. | |
| Bacteria OTU richness OTUs | O | 726 (84) | 756 (57) | 663 (171) | 623 (73) | 680 (87) | 727 (117) | n.s. | |
| | A | 695 (50) a | 655 (70) ab | 596 (89) abc | 548 (89) bc | 565 (50) bc | 533 (58) c | 4.49 | 0.004 |
| | B | 612 (72) a | 620 (66) a | 607 (33) a | 637 (20) a | 565 (56) ab | 512 (39) b | 6.88 | <0.001 |
| Bacteria Shannon | O | 5.55 (0.24) | 5.60 (0.18) | 5.34 (0.40) | 5.33 (0.19) | 5.41 (0.25) | 5.57 (0.23) | n.s. | |
| | A | 5.38 (0.15) a | 5.23 (0.16) ab | 5.15 (0.22) ab | 4.91 (0.34) b | 4.91 (0.14) b | 4.99 (0.20) b | 4.83 | 0.003 |
| | B | 5.08 (0.25) a | 5.05 (0.19) ab | 5.13 (0.09) a | 5.14 (0.09) a | 4.92 (0.19) ab | 4.79 (0.15) b | 5.2 | 0.001 |
| Fungal 18S rRNA gene $10^7$ copies g$^{-1}$ | O | 129 (110) | 58 (85) | 147 (89) | 52 (51) | 13 (7) | 60 (78) | n.s. | |
| | A | 13.1 (0.6) a | 7.4 (0.5) ab | 41.9 (5.5) a | 9.1 (0.6) a | 11.2 (1) a | 1.3 (0.1) b | 5.4 | 0.001 |
| | B | 3.76 (0.35) | 0.28 (0.02) | 1.74 (0.15) | 1.17 (0.15) | 5.2 (0.53) | 2.43 (0.21) | n.s. | |
| Bacterial 16S rRNA gene $10^9$ copies g$^{-1}$ | O | 36.7 (1.9) a | 27.8 (5) ab | 22.6 (2.5) ab | 6.2 (1.6) c | 9 (1.4) bc | 8.6 (1.4) bc | 9.51 | <0.001 |
| | A | 16.6 (0.6) ab | 29.2 (2.2) a | 20.1 (2.2) ab | 7.8 (1.2) bc | 4.2 (0.4) c | 4.5 (1.4) c | 11.7 | <0.001 |
| | B | 9.65 (1.68) a | 7.83 (2.26) ab | 1.84 (0.49) c | 1.05 (0.16) bc | 3.98 (0.75) abc | 4.82 (0.84) abc | 3.99 | 0.007 |
| Fungi-to-bacteria (F:B) copy:copy | O | 0.035 (0.027) | 0.028 (0.046) | 0.069 (0.034) | 0.098 (0.118) | 0.017 (0.008) | 0.055 (0.059) | n.s. | |
| | A | 0.008 (0.003) ab | 0.003 (0.002) a | 0.016 (0.017) ab | 0.011 (0.004) ab | 0.025 (0.021) b | 0.007 (0.01) ab | 3.56 | 0.014 |
| | B | 0.003 (0.002) ab | 0.001 (0.001) a | 0.008 (0.007) b | 0.018 (0.031) b | 0.011 (0.009) b | 0.006 (0.006) b | 3.26 | 0.018 |

The most abundant fungal taxa across all studied sites were ectomycorrhizal *Tylospora* sp., *Russula* sp., *Piloderma* sp. and *Amanita* sp., together with Mortierellales moulds and Helotiales (Table 4 and Table S2). Ectomycorrhizal fungi formed 45–59% and 42–76% of the fungal amplicons in O and A horizons, respectively, but only 20–51% in B horizons, comparable among the sites in either of the horizons (Table 4). There was a significant effect of the time since disturbance on the fungal communities' composition in A and B horizons (Table S3), but post hoc site pairwise comparison was insignificant for most tested fungal genera and orders (Table S2). In O horizons, the fungal communities differed significantly between the O horizon types (Table S4).

The bacterial communities were generally dominated by Acidobacteriae (~40%), and the other classes found in high relative abundance across all sites and horizons were Alpha- and Gammaproteobacteria, Actinobacteria, Verrucomicrobiae and Planctomycetes (Table 5 and Table S5). In O horizons, Bacteroidia also occurred in considerable numbers, while Nitrososphaeria and Ktedonobacteria were numerous in A and B horizons (Table 5 and Table S5). Bacterial communities differed among sites but only in A and B horizons, while the type of organic horizon importantly shaped the communities in O horizons (Tables S3 and S6).

**Table 4.** Site differences in the respective horizons for the most abundant ectomycorrhizal genera/orders and all ectomycorrhizal fungi summed. Values represent the mean of relative abundance (in %) with standard deviation in parentheses. Small letters denote differences significant at $p < 0.05$. For a complete list of all of the most abundant fungal genera and orders, see Table S2.

| | | | | | | | | ANOVA Sites | |
| --- | --- | --- | --- | --- | --- | --- | --- | --- | --- |
| | Horizon | 16 Years | 36 Years | 100 Years | 110 Years | 130 Years | 160 Years | F | p |
| *Amanita* | O | 2.3 (4.1) | 0.8 (1.1) | <0.1 | 2.7 (6) | <0.1 | 3.4 (7.6) | n.s. | |
| | A | 10.1 (22.1) | 0.5 (1.1) | <0.1 | 6.9 (12.2) | 7.4 (8.3) | 11.7 (23.1) | n.s. | |
| | B | 15.4 (26.5) | <0.1 | <0.1 | 1.2 (2.3) | <0.1 | <0.1 | n.s. | |
| *Boletus* | O | <0.1 | <0.1 | 0.3 (0.5) | 0.4 (0.5) | <0.1 | <0.1 | n.s. | |
| | A | <0.1 c | <0.1 c | 6.6 (3.8) a | 2.1 (2.7) b | <0.1 c | <0.1 c | 11.7 | <0.001 |
| | B | <0.1 b | <0.1 b | 16.8 (16) a | 2.8 (4.2) ab | <0.1 b | <0.1 b | 5.8 | 0.001 |
| Cantharellales | O | 3.6 (7.1) | 0.3 (0.1) | 0.4 (1) | 2.5 (4.5) | 0.3 (0.5) | <0.1 | n.s. | |
| | A | 4.2 (9.1) ab | <0.1 b | <0.1 b | 24.5 (24.2) a | 2 (4.5) b | <0.1 b | 4.6 | 0.006 |
| | B | 0.4 (0.8) a | <0.1 a | <0.1 a | 17.8 (24.1) a | 28.2 (48.5) a | <0.1 a | 3.0 | 0.029 |
| *Hygrophorus* | O | 11.3 (8.7) | <0.1 | 4.8 (5.6) | 1.2 (1.9) | <0.1 | 4.1 (7.2) | n.s. | |
| | A | 10.8 (20.2) | 0.2 (0.4) | 0.8 (1.1) | <0.1 | 0.2 (0.4) | 0.3 (0.6) | n.s. | |
| | B | <0.1 | <0.1 | <0.1 | <0.1 | <0.1 | <0.1 | n.s. | |
| *Meliniomyces* | O | 0.2 (0.1) c | 2.8 (0.4) ab | 0.7 (0.7) bc | 4 (1.7) a | 1.1 (0.8) bc | 2.4 (1) ab | 12.6 | <0.001 |
| | A | 0.1 (0.3) | <0.1 | 0.4 (0.5) | 0.4 (0.4) | 0.7 (0.8) | 1.6 (2.7) | n.s. | |
| | B | <0.1 | <0.1 | <0.1 | <0.1 | 6.1 (10.6) | 0.2 (0.1) | n.s. | |
| *Piloderma* | O | 4.6 (8.5) | <0.1 | 14.1 (17.1) | 11.1 (9.6) | 2 (4) | 10.1 (12.7) | n.s. | |
| | A | <0.1 | <0.1 | 8.3 (11.5) | 10.2 (16.6) | 13.1 (17.3) | 2.8 (3.9) | n.s. | |
| | B | <0.1 a | <0.1 a | 0.4 (0.5) a | <0.1 a | 0.2 (0.1) a | <0.1 a | 2.7 | 0.045 |
| *Russula* | O | <0.1 | 9.6 (13.5) | 12.7 (27.7) | 9.1 (17.8) | 21.4 (24.3) | 10.7 (23.8) | n.s. | |
| | A | <0.1 | 3.6 (7.8) | 30.6 (43.1) | 4.1 (9.2) | 9.6 (7.6) | 14.5 (21.5) | n.s. | |
| | B | <0.1 | <0.1 | 0.5 (0.5) | <0.1 | <0.1 | 5.6 (9.7) | n.s. | |
| *Thelephorales* | O | 1.3 (1.4) | 0.4 (0.2) | 1.1 (2) | 0.3 (0.3) | 1.1 (2.1) | 0.6 (0.5) | n.s. | |
| | A | 0.3 (0.7) | 0.2 (0.2) | <0.1 | <0.1 | 0.7 (1) | 4.2 (7.8) | n.s. | |
| | B | <0.1 | 2.1 (3) | 2.1 (3.6) | <0.1 | <0.1 | 0.1 (0.1) | n.s. | |
| *Tylospora* | O | 32.6 (18.2) | 48.4 (13.9) | 16.3 (15.1) | 20.3 (20.3) | 22.1 (27.5) | 17.2 (22.6) | n.s. | |
| | A | 48.6 (28.9) | 59.3 (33) | 25.9 (22.4) | 11.9 (16) | 25.9 (23.9) | 16.8 (11.7) | n.s. | |
| | B | 20.9 (23) | 14.8 (26.4) | 21.9 (24.2) | 20.7 (21.4) | 0.1 (0.1) | 24.2 (33.4) | n.s. | |
| *Wilcoxina* | O | <0.1 | <0.1 | <0.1 | <0.1 | <0.1 | <0.1 | n.s. | |
| | A | 5.7 (10.4) | 4.4 (8.7) | <0.1 | <0.1 | <0.1 | <0.1 | n.s. | |
| | B | 14.8 (20) | 4.6 (8.9) | 0.9 (2.1) | 2.2 (3.2) | <0.1 | <0.1 | n.s. | |
| All ectomycorrhizal summed | O | 52.1 (26.8) | 59.3 (0.7) | 53.9 (32.3) | 45.2 (13.3) | 46.8 (23.7) | 48.4 (27.2) | n.s. | |
| | A | 75.6 (12.6) | 68.4 (37.2) | 72.3 (6.5) | 42.2 (23.2) | 58.9 (16.7) | 46.8 (33.5) | n.s. | |
| | B | 51.1 (17.1) | 19.5 (24.6) | 41 (23.6) | 33.8 (28.8) | 28.9 (48.2) | 30.3 (30.9) | n.s. | |

### 3.4. Factors Shaping Microbial Communities in Respective Horizons

3.4.1. Surface Organic Horizons

Bacterial and fungal communities in O horizons were significantly driven by the O horizon type (28.8% and 20.4% of explained variability by PERMANOVA for bacteria and fungi, respectively; both $p < 0.001$; Table S3). The distance-based redundancy analysis (db-RDA) ordination clearly separated microorganisms associated with O horizon types T, L + F and H (Figures 2A and 3A), which means that communities in grass-litter-derived T-type O horizons present only in the gaps differed from microbial assemblages in the mature stands. Additionally, the microbiome composition differed between the upper fresh and partly decomposed litter in the L + F layer from the humified H layer within the respective mature forest sites. Different key factors drove the differences in fungi and bacteria. The CEC and $C_{tot}{:}N_{tot}$ were the most influential factors for the fungal community composition in O horizons (Table S7). The fungal communities of litter layers (L + F) with the highest $C_{tot}{:}N_{tot}$ ratios were separated along the *x* axis from fungi in humic horizons (H) with low-to-intermediate $C_{tot}{:}N_{tot}$ ratios but the highest CEC (Figure 2A). The *y* axis then distinguished fungal communities of T-type organic horizons of 16- and 36-year sites having

the lowest $C_{tot}$:$N_{tot}$ ratios from those in the mature stands (Figure 2A). The low C:N T-type organic horizon of the two sites with the shortest time since disturbance was preferred by *Tylospora asterophora* and *Mortierella humilis* (Figure S1A). *Tylospora fibrillosa*, *Amphinema byssoides*, *Cladophialophora* sp. and *Auricularia* sp. were commonly found in the litter (L + F) layer in mature stands. Aside from *Piloderma sphaerosporum*, other ectomycorrhizal genera such as *Russula*, *Amanita* and *Hygrophorus* also preferred the H horizons. The humus layer was further favoured by *Mortierella* sp. and *Cryptococcus terricola*.

**Table 5.** Bacterial classes with significant site differences in the respective horizons. Values represent the mean of relative abundance (in %) with standard deviation in parentheses. Small letters denote differences significant at *p* < 0.05. For a complete list of all of the most abundant bacterial classes, see Table S5.

| | Horizon | 16 Years | 36 Years | 100 Years | 110 Years | 130 Years | 160 Years | ANOVA Sites F | ANOVA Sites p |
|---|---|---|---|---|---|---|---|---|---|
| Acidimicrobiia | O | 1.20 (0.35) c | 1.54 (0.72) bc | 2.08 (0.58) abc | 3.65 (1.25) a | 2.92 (1.11) ab | 3.10 (1.34) ab | 5.99 | <0.001 |
| | A | 0.95 (0.41) ab | 0.62 (0.17) b | 1.54 (1.09) ab | 1.91 (1.06) a | 1.70 (0.41) a | 1.41 (0.69) ab | 3.57 | 0.013 |
| | B | 1.35 (0.45) ab | 0.96 (0.28) b | 1.46 (0.24) ab | 1.75 (0.29) a | 1.38 (0.23) ab | 1.84 (0.44) a | 5.64 | <0.001 |
| Actinobacteria | O | 3.97 (1.03) | 4.14 (2.6) | 6.08 (2.22) | 8.43 (3.21) | 6.79 (2.59) | 6.65 (3.81) | n.s. | |
| | A | 2.75 (1.28) bc | 1.85 (0.64) c | 6.50 (2.35) a | 4.73 (2.14) ab | 4.79 (1.18) ab | 4.17 (1.85) abc | 5.61 | 0.001 |
| | B | 1.54 (1.11) bc | 0.51 (0.31) c | 1.29 (0.72) bc | 1.76 (0.82) bc | 2.91 (2.18) b | 5.24 (0.92) a | 12.3 | <0.001 |
| AD3 | O | 0.11 (0.04) | 0.20 (0.16) | 0.14 (0.13) | 0.12 (0.05) | 0.19 (0.14) | 0.22 (0.22) | n.s. | |
| | A | 1.25 (0.78) a | 0.99 (0.77) a | 0.38 (0.33) a | 0.92 (1.19) a | 0.23 (0.17) a | 0.19 (0.16) a | 3.01 | 0.028 |
| | B | 4.36 (2.16) cd | 3.14 (0.98) bd | 6.82 (1.51) c | 3.09 (1.17) bd | 1.67 (1.0) ab | 1.19 (0.77) a | 13.6 | <0.001 |
| Alphaproteobacteria | O | 15.6 (2.5) a | 13.8 (2.2) a | 13.7 (3.2) a | 17.3 (1.8) a | 17.9 (2.2) a | 17.6 (2.5) a | 2.87 | 0.030 |
| | A | 11.4 (5.2) b | 9.5 (4.5) b | 17.5 (2.5) ab | 15.2 (2.8) ab | 18.7 (3.9) a | 20.9 (3.1) a | 6.83 | <0.001 |
| | B | 9.7 (3.4) bc | 7.6 (1.4) c | 9.8 (2.5) bc | 10.8 (3.2) bc | 13.3 (3.9) ab | 19.5 (5.6) a | 8.10 | <0.001 |
| Bacteroidia | O | 7.51 (3.12) | 7.04 (1.21) | 7.93 (5.26) | 5.53 (3.03) | 5.96 (3.18) | 6.63 (3.40) | n.s. | |
| | A | 2.22 (1.23) a | 2.40 (0.85) a | 3.00 (1.24) a | 1.35 (0.97) a | 1.22 (0.26) a | 1.41 (0.47) a | 3.14 | 0.024 |
| | B | 0.61 (0.21) | 0.82 (0.44) | 1.26 (0.86) | 0.79 (0.42) | 0.88 (0.50) | 0.66 (0.20) | n.s. | |
| Chlamydiae | O | 0.65 (0.26) | 0.86 (0.59) | 0.88 (0.60) | 1.5 (1.43) | 0.86 (0.40) | 1.34 (1.06) | n.s. | |
| | A | 1.33 (0.32) | 1.27 (0.26) | 0.85 (0.34) | 1.68 (0.79) | 0.9 (0.27) | 1.72 (0.93) | n.s. | |
| | B | 1.42 (0.23) b | 2.02 (0.71) ab | 1.36 (0.48) b | 2.75 (0.50) a | 1.7 (0.53) b | 1.29 (0.38) b | 6.72 | <0.001 |
| Fimbriimonadia | O | 0.51 (0.17) b | 0.37 (0.02) ab | 0.27 (0.14) a | 0.22 (0.09) a | 0.25 (0.11) a | 0.33 (0.09) ab | 3.87 | 0.009 |
| | A | 0.22 (0.15) b | 0.14 (0.06) ab | <0.1 ab | <0.1 a | <0.1 a | <0.1 a | 4.41 | 0.005 |
| | B | 0.1 (0.03) bc | 0.14 (0.04) c | 0.15 (0.04) c | <0.1 bc | <0.1 ab | <0.1 a | 8.68 | <0.001 |
| Gammaproteobacteria | O | 13.12 (3.56) | 11.41 (5.46) | 9.66 (3.83) | 10.21 (5.62) | 9.12 (3.92) | 8.60 (3.10) | n.s. | |
| | A | 5.37 (0.76) | 4.96 (1.05) | 6.77 (1.90) | 7.39 (5.64) | 3.67 (1.83) | 4.01 (1.03) | n.s. | |
| | B | 6.76 (1.37) b | 8.08 (2.03) b | 5.51 (0.82) b | 5.31 (1.24) b | 5.38 (1.96) b | 2.97 (1.00) a | 8.82 | <0.001 |
| Gemmatimonadetes | O | 0.65 (0.26) b | 0.47 (0.14) ab | 0.32 (0.19) ab | 0.20 (0.11) a | 0.32 (0.15) ab | 0.36 (0.20) ab | 4.06 | 0.007 |
| | A | 1.50 (0.28) c | 1.27 (0.53) bc | 0.75 (0.12) abc | 0.79 (0.71) ab | 0.74 (0.25) ab | 0.50 (0.24) a | 5.17 | 0.002 |
| | B | 1.80 (0.46) cd | 1.31 (0.33) bcd | 1.99 (0.57) d | 1.09 (0.43) bc | 0.97 (0.32) b | 3.01 (0.68) a | 14.0 | <0.001 |
| Ktedonobacteria | O | 0.20 (0.27) | 0.47 (0.56) | 0.16 (0.23) | <0.1 | <0.1 | <0.1 | n.s. | |
| | A | 5.20 (1.89) b | 6.15 (3.1) b | 2.27 (2.06) ab | 3.15 (4.58) a | 0.21 (0.30) a | <0.1 a | 6.81 | <0.001 |
| | B | 7.31 (1.62) | 11.02 (5.72) | 9.46 (3.31) | 10.62 (3.84) | 7.03 (4.35) | 4.88 (3.32) | n.s. | |
| Myxococcia | O | 0.49 (0.11) | 0.23 (0.13) | 0.31 (0.18) | 0.31 (0.13) | 0.29 (0.07) | 0.45 (0.25) | n.s. | |
| | A | 0.26 (0.06) | 0.28 (0.13) | 0.17 (0.06) | 0.20 (0.25) | 0.16 (0.05) | 0.15 (0.06) | n.s. | |
| | B | 0.55 (0.27) c | 0.46 (0.09) bc | 0.28 (0.13) ab | 0.21 (0.12) a | 0.22 (0.07) a | 0.14 (0.04) a | 9.53 | <0.001 |
| Nitrososphaeria | O | 0.18 (0.12) | 0.26 (0.19) | 0.20 (0.25) | 0.16 (0.18) | 0.24 (0.36) | 0.23 (0.27) | n.s. | |
| | A | 3.45 (1.58) bc | 4.07 (1.58) c | 1.16 (0.4) ab | 1.37 (1.27) a | 1.28 (0.39) a | 1.34 (0.79) a | 7.28 | <0.001 |
| | B | 5.61 (1.44) ab | 8.55 (1.38) b | 5.28 (1.67) ab | 5.12 (3.46) ab | 3.01 (1.49) a | 4.15 (1.52) a | 5.35 | 0.001 |
| RCP2-54 | O | 1.70 (1.25) | 1.81 (1.80) | 1.21 (1.21) | 1.73 (1.66) | 1.54 (1.45) | 2.09 (1.77) | n.s. | |
| | A | 4.54 (1.26) ab | 3.65 (1.25) b | 4.29 (1.59) ab | 5.30 (2.90) ab | 6.31 (1.22) ab | 7.69 (3.16) a | 2.87 | 0.034 |
| | B | 2.59 (0.78) bcd | 2.09 (0.71) cd | 1.69 (0.27) d | 3.17 (0.83) abc | 3.70 (1.40) ab | 4.57 (0.37) a | 10.2 | <0.001 |
| Thermoleophilia | O | 1.05 (0.38) | 1.37 (1.06) | 1.60 (0.86) | 2.84 (1.04) | 2.24 (1.24) | 2.23 (1.48) | n.s. | |
| | A | 0.60 (0.29) bc | 0.31 (0.14) c | 0.96 (0.41) ab | 1.29 (0.62) a | 0.90 (0.40) ab | 0.55 (0.13) bc | 6.18 | <0.001 |
| | B | 0.62 (0.3) bc | 0.50 (0.17) c | 1.16 (0.32) a | 0.93 (0.32) ab | 0.80 (0.12) abc | 0.79 (0.18) abc | 5.32 | 0.001 |
| Verrucomicrobiae | O | 5.66 (1.19) | 5.88 (1.27) | 6.43 (1.93) | 5.00 (1.41) | 6.01 (1.61) | 6.57 (2.24) | n.s. | |
| | A | 4.89 (0.77) b | 5.07 (1.57) b | 5.32 (0.28) b | 4.25 (0.87) ab | 3.16 (0.49) a | 4.39 (0.95) ab | 3.86 | 0.010 |
| | B | 3.03 (0.92) ab | 2.56 (0.86) ab | 3.94 (1.17) b | 2.85 (1.36) ab | 3.67 (0.63) ab | 2.19 (0.89) a | 2.75 | 0.037 |

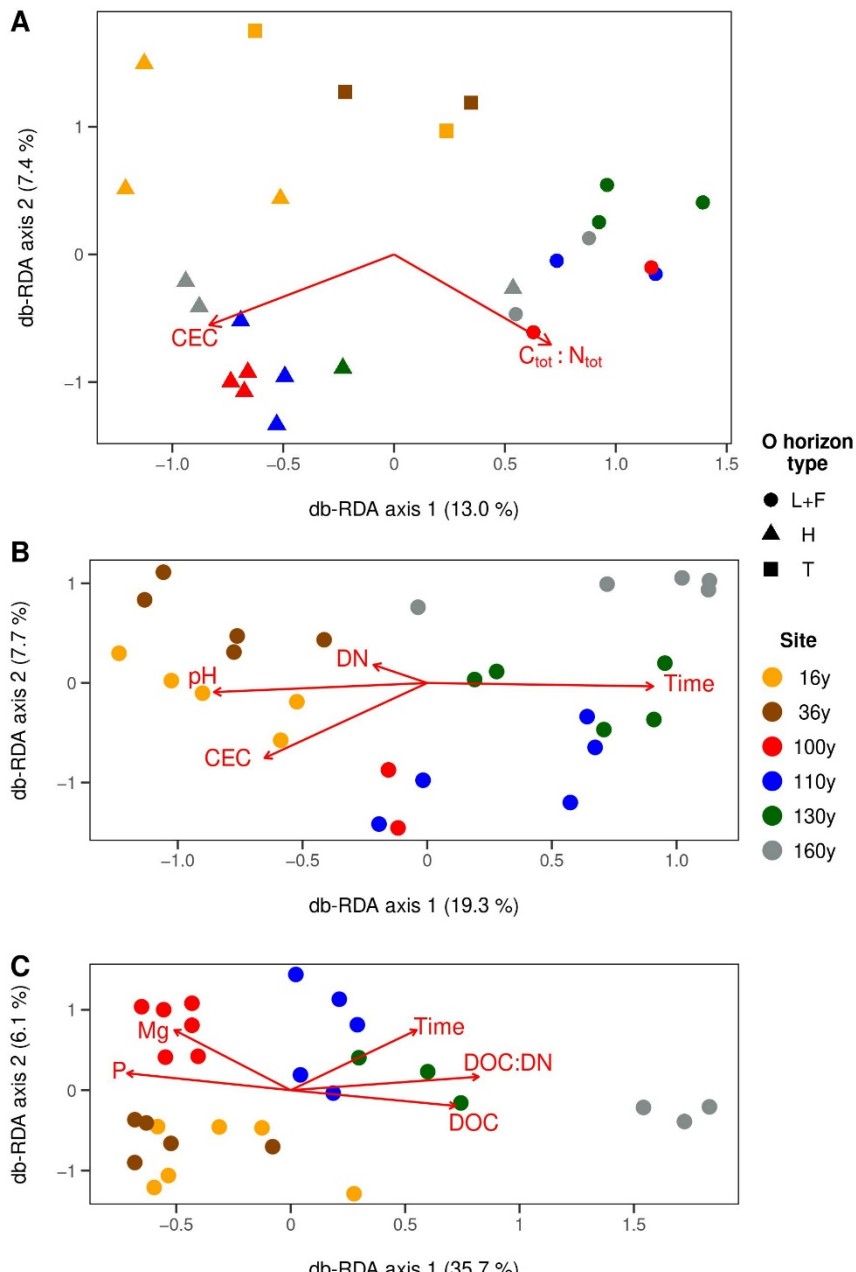

**Figure 2.** Distance-based redundancy analysis (db-RDA) ordinations built on variables that explained most of the variability in the fungal community composition in (**A**) O horizons, (**B**) A horizons and (**C**) B horizons. Symbol colours represent the site. Symbol shapes in panel (**A**) denote the O horizon type. The axes are rescaled according to the explained variability.

Bacteria were the most influenced by pH and by $C_{tot}:N_{tot}$ in O horizons. The position on the *x* axis separated samples according to pH, where the communities of L + F and T types were agglomerated at high-to-intermediate pH levels, while H-type horizons were located at the lowest end of the pH gradient in the db-RDA ordination plot (Figure 3A). The differences between O-horizon-type-associated bacterial communities further stemmed from differences in $C_{tot}:N_{tot}$. The youngest 16- and 36-year sites with T-type horizons with the lowest $C_{tot}:N_{tot}$ were divided from the mature forest stands laying along the $C_{tot}:N_{tot}$ continuum. The canonical correspondence analysis (CCA) plot suggests that the more acidic conditions in H-type horizons were preferred by Acidobacteriae and Planctomycetes, while OTUs belonging to Bacteroidia and Gammaproteobacteria were inclined to favour a higher pH (Figure S2A). The most abundant OTUs were clustered closer to L + F- and

H-type organic horizons, which implies the preference of these bacteria that occur in high relative abundance across all sites for organic material with a higher C:N ratio.

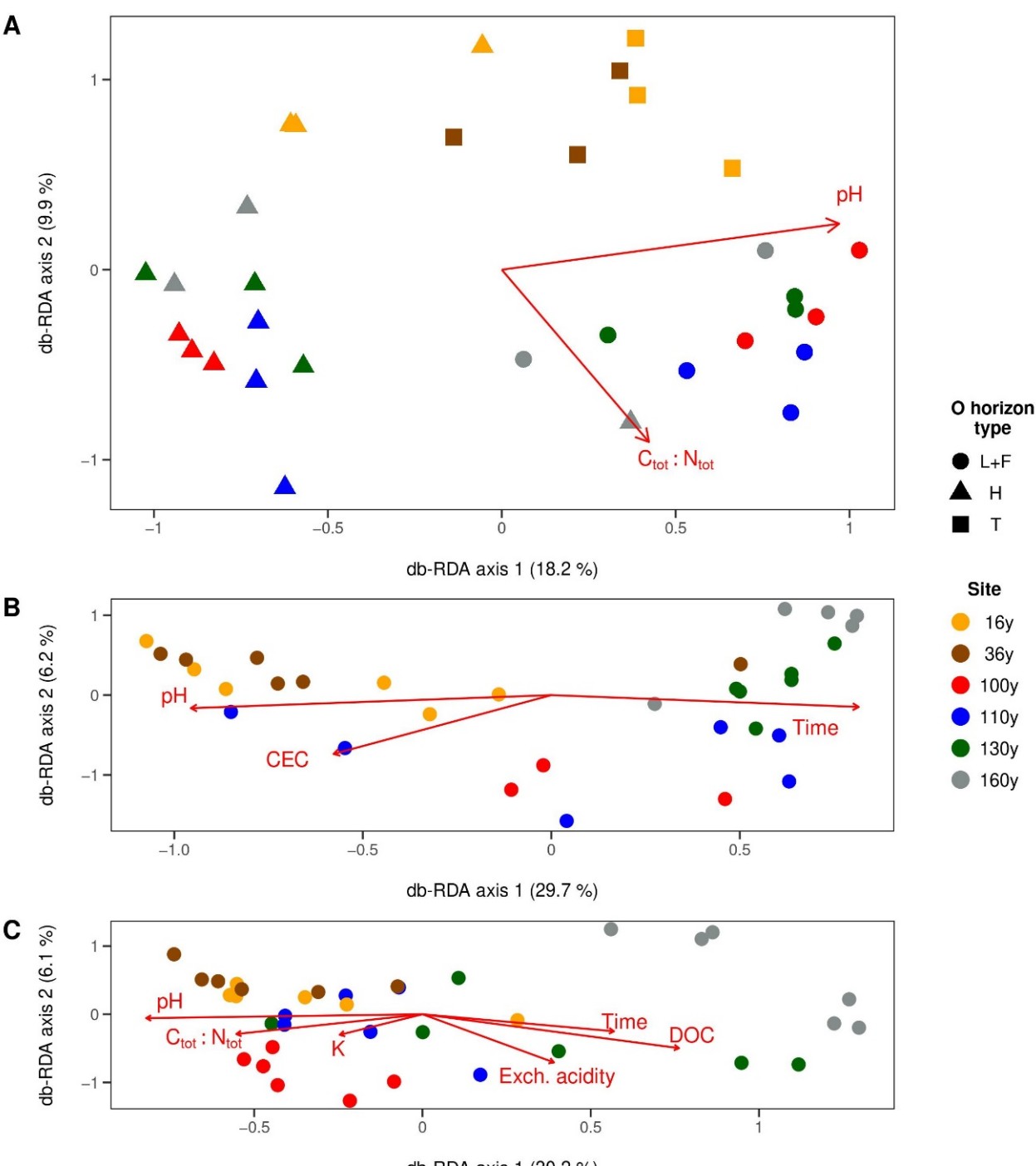

**Figure 3.** Distance-based redundancy analysis (db-RDA) ordinations built based on variables that explained most of the variability in the bacterial community composition in (**A**) O horizons, (**B**) A horizons and (**C**) B horizons. Symbol colours represent the site. Symbol shapes in panel (**A**) denote the O horizon type. The axes are rescaled according to the explained variability.

### 3.4.2. Upper Mineral Horizons

The microbiome composition in A horizons was mostly determined by the time since the last disturbance connected with the decreasing soil pH and CEC that were significantly

lower in 130- and 160-year sites than in the other sites (Table 2). The biggest part of the variability in the fungal community was explained by the time since disturbance (19.3%, Table S7). The db-RDA ordination plot showed a clear separation of the 16- and 36-year sites from the mature stand sites and a continuous distribution of old-growth forest sites along the time gradient (Figure 2B). Soil pH was the dominant influential variable for bacterial A horizon communities (29.7% explained variability; Table S7). In the db-RDA plot, the 16- and 36-year sites with the highest pH were separated from the mature stands (Figure 3B). CCA suggested a preference of Ktedonobacteria and Bacteroidia for soils with higher pH, while Alphaproteobacteria inclined to occur in higher abundance in more acidic soils with a longer time since disturbance (Figure S2B).

### 3.4.3. Spodic Horizons

The fungal communities in B horizons were mostly driven by the gradient of the quality and the amount of incoming dissolved organic matter represented by the DOC:DN and DOC concentrations, respectively (Figure 2C, Table S7). The 16-, 36- and 100-year sites with a rather low DOC and a DOC:DN of <20 clustered at the left side of the x axis, which copied this gradient, the 160-year site with the highest DOC and DOC:DN was placed at the opposite side and the 110- and 130-year sites were intermediate (Figure 2C). The contribution of the other significant factors to the explained variability was low.

For bacteria, likewise in the other horizons, the superior factor was pH, which explained >30% of the variability, while the other factors contributed considerably less to the total explained variability (Figure 3C, Table S7). There was a sign of the pH gradient, time since disturbance and DOC concentration from the left to the right side of the x axis. However, the samples from the respective sites overlayed each other (Figure 3C). CCA suggested that Ktedonobacteria and Nitrososphaeria prefer soils with a higher pH (Figure S2C).

### 3.5. Microbial Groups Potentially Participating in Nitrification, N Fixation and Nitrate Respiration

Potential nitrifiers made up less than 0.5% of prokaryotes in O horizons across all sites. However, they reached up to 1.7% and almost 3% in A and B horizons, respectively, in the 16- and 36-year sites, particularly due to a high relative abundance of Nitrososphaeria (namely Nitrosotaleaceae; Figure 4A), while their abundance was much lower in old-growth forests. The relative abundance of potential nitrifiers in A and B horizons was negatively correlated with DOC:DN ($r = -0.46$, $p = 0.008$ and $r = -0.398$, $p = 0.018$, respectively; Figure S3A).

The relative abundance of OTUs with the potential capability to fix N was similar in O horizons across all sites. In A and B horizons, the relative abundance of potential fixators gradually increased from both regenerating 16- and 36-year sites along the time since disturbance up to the maximum at the 160-year site (Figure 4B). In A and B horizons, the relative abundance of potential N fixators was positively correlated to DOC:DN ($r = 0.57$, $p < 0.001$ and, $r = 0.46$, $p = 0.004$, respectively; Figure S3B).

The relative abundance of OTUs that can potentially use nitrate as a terminal acceptor of electrons was positively correlated to DOC:DN across all horizons ($r = 0.39$, $p = 0.026$; $r = 0.51$, $p = 0.003$; $r = 0.64$, $p < 0.001$ in O, A and B horizons, respectively; Figure S3C). Potential nitrate respirers were similarly present in O horizons across the sites. However, they were significantly less abundant in 16- and 36-year sites compared to mature forests in A horizons and the least abundant in the 36-year site and further gradually increased with time since the last disturbance in B horizons (Figure 4C).

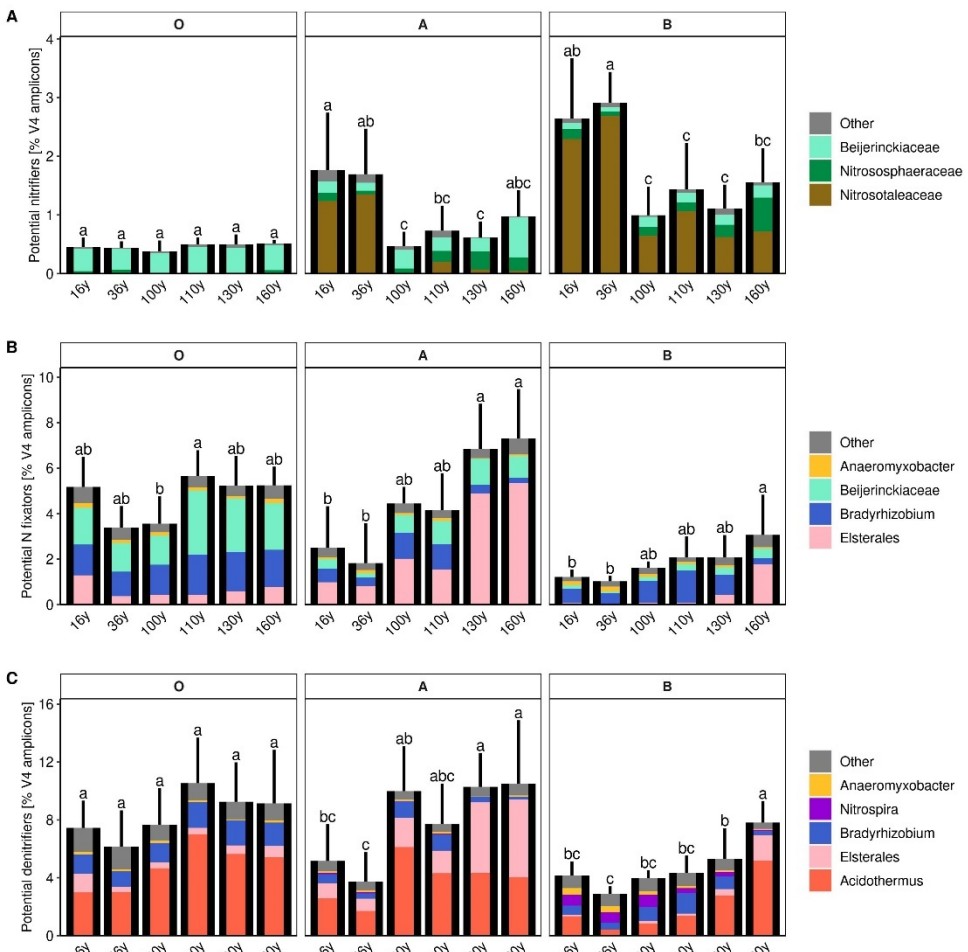

**Figure 4.** Relative abundance of microbes potentially capable of (**A**) nitrification, (**B**) N fixation and (**C**) denitrification in the respective sites and horizons. The outer black columns and error bars represent the mean and SD of all selected OTUs, respectively. Small letters denote differences significant at $p < 0.05$ between the summed relative abundance of the respective groups. The coloured inner columns show the taxonomic classification of included OTUs.

## 4. Discussion

### 4.1. Vegetation and Soil Chemistry along the Chronosequence

The studied chronosequence consisted of six primeval Norway spruce forest sites located in the most preserved core area of the Calimani National Park, where the ecosystems naturally evolved since the last stand-replacing disturbance without any signs of direct human intervention [50]. The two sites with the shortest period of post-disturbance development (16 and 36 years) represented still open, but gradually closing gaps covered by grass vegetation. The remaining four sites were already compact-canopy forest stands 100–160 years since the last disturbance. The canopy opening and expansion of the understorey (in particular *Calamagrostis villosa*, but also *Nardus stricta*, *Avenella flexuosa* and *Athyrium distentifolium*) changed the quality of incoming litter and other plant-derived C inputs into the soil and enhanced the decomposition of organic matter accumulated during the previous phase of the old-growth forest. The typical spruce forest mor layer consisting of decomposing needles in thick L, F and especially H horizons was replaced in these gaps by a comparatively thin T horizon originating from tissues of living graminoids and their residues [57]. The organic soil horizons in both gradually closing gaps had lower $C_{tot}:N_{tot}$ as well as DOC:DN compared to the old-growth forests. The change in the nutritional quality of incoming material also projected deeper into upper mineral and spodic horizons. However, unlike the distinctly different C:N ratios in O horizons between

gaps and closed forests, the C:N ratio in the deeper horizons rather gradually increased along the chronosequence from the lowest at the 36-year site to the largest at the 160-year site. Soil pH followed an opposite gradient in these horizons, being the highest in the gaps and gradually decreasing with the time since disturbance.

### 4.2. General Patterns in the Soil Microbiome Composition

All the sites were inhabited by soil microbiomes typical for spruce forests. Fungal communities were mostly formed by ectomycorrhizal fungi, followed by moulds (namely Mortierelalles) and Helotiales, a diverse order of litter saprobes and root symbionts, groups typically highly abundant in Norway spruce forests [23,26]. Bacterial communities were dominated by groups tolerant to low pH and capable of degrading complex organic compounds: Acidobacteriae, Actinobacteria (namely *Acidothermus* sp.) and Verrucomicrobiae [88–90].

Consistent with our hypothesis (1), the composition of the soil microbial community in the respective sites reflected the time since disturbance. Compared to changes in the soil chemistry observed along the chronosequence, the shifts in the microbiome were more pronounced and occurred through the whole soil profile from O to B horizons. Microbial communities in the O horizons were crucially shaped by the quality of aboveground plant litter and its degradability, specifically by the selection of species capable of decomposing certain compounds and by the amount of nutrients released during this decomposition [19,23,91]. Therefore, the communities in O horizons of gaps with high grass coverage differed from assemblages in spruce-litter-derived L + F and H layers of the O horizon in old-growth forests. While spruce litter hosted a specific microbial community that was able to enzymatically degrade recalcitrant compounds and tolerate decomposition-related acidification [92–94], grass litter is more easily decomposable, containing a high amount of available nutrients [28].

### 4.3. Soil Bacterial Communities

The soil microbial communities in gaps were typical by a larger contribution of bacteria across all horizons compared to old-growth forest microbiomes. The bacterial communities in gaps were further of a significantly different composition and of slightly higher richness than those in the old-growth forests. The soil pH gradient was the most important factor shaping bacterial communities in all horizons. Due to the partial release of acidity-driven stress and higher N availability in the soils in gaps compared to mature forests, we observed a lower relative abundance of acidotolerant Acidimicrobiia and Actinobacteria (mostly *Acidothermus* sp.) [95,96] but a higher contribution of Bacteroidia and Gemmatimonadetes that do not prosper in acidic conditions [97]. However, the majority of other bacterial classes were similarly abundant as in the old-growth forests. The strongly acidic soils with a high C:N ratio of old-growth forests were particularly preferred by Acidobacteriae, Actinobacteria, Planctomycetes and Verrucomicrobiae, classes with members capable of degrading complex organic compounds and in many cases tolerant to low pH [88–90,98]. Although these oligotrophic bacteria made a considerable part of the bacterial community in the two gaps as well, higher N availability in combination with greater substrate degradability enhanced the competitiveness and representation of copiotrophs, which projected to differences in the bacterial community composition and its richness, as anticipated in hypothesis (1).

The gradient of decreasing soil N availability, being the highest in gap sites and continuously declining during forest succession, was followed by altered abundances of various bacterial groups involved in specific processes of N transformation. In N-richer soils in gaps, potential nitrifiers were twofold more abundant in A and B horizons compared to the closed forests. Their higher abundance in these soils with a low C:N ratio might stem not only directly from a high N availability but also from a low DOC availability that limits the competitiveness of heterotrophs [99]. On contrary, bacteria potentially capable of $N_2$ fixation and nitrate respiration were the least numerous in gaps,

and their relative abundance continually increased with an increasing C:N ratio along the chronosequence. Both respiratory pathways that use nitrate as an electron acceptor, denitrification and dissimilatory nitrate reduction to ammonium, can occur in microsites with $O_2$ scarcity and the presence of nitrates, but also sufficient availability of degradable sources of C [100,101], which explains their increasing relative abundance with relatively decreasing N but increasing C availability. The dependence of their relative abundance on the C:N ratio was generally more pronounced down the soil profile. That further indicates the possible importance of the organic C supply to deeper soil horizons, which is needed to reduce nitrates. N fixation is a significant source of N in unpolluted N-limited forests [102], and our results suggest that its importance increases with proceeding forest succession and N scarcity due to its gradual immobilization in tree biomass and stabilized soil organic matter [27].

### 4.4. Soil Fungal Communities

The disturbance projected to the composition of fungal communities as well, although we found no systematic differences in the relative abundance of fungal taxa, nor 18S rRNA gene abundance and diversity measures between the sites along the chronosequence. In O and A horizons, we observed a similar pattern as in the case of bacteria. The fungal communities in organic T-type horizons with a lower C:N ratio in gaps clearly differed from those in the litter of old-growth forests, while in A horizons, a continuous distribution of site-specific fungal communities along the time gradient in the ordination suggests their development, which mirrors the forest succession. The succession with stand age is a typical feature of the ectomycorrhizal community [20,38,103]. Accordingly, the ectomycorrhizal community in gaps was strongly prevailed by *Tylospora* sp., and *Wilcoxina* sp. was considerably present only in these sites. Both these genera are typical for early successional stages, while *Russula* sp., which was missing in the 16-year site, occurs rather in closed forests [38,43,44]. However, the relative abundance of all ectomycorrhizal fungi in gaps was similar as in old-growth forests, which contradicts our hypothesis (1). Due to a dependence of ectomycorrhizal fungi on living tree roots that mediate the supply of tree photosynthates [104,105], we expected a rapid reduction in their presence at young sites being 16–36 years after disturbance. However, reports of such a vast decline come from large-scale, clear-cut or naturally disturbed sites [38,43,106,107]. However, the size of gaps in our study was smaller (<0.1 ha), and although there were no living trees within the sampling site, the roots extending from the surrounding forest might have provided a niche for the survival of ectomycorrhizal fungi [108]. The majority of ectomycorrhizal sequences in gaps belonged to *Tylospora* sp. Molecular analyses suggested that a member of this genus, *T. fibrillose*, potentially possesses genes for enzymatic breakdown of complex organic matter [109]. This potential ability and the fact that *Tylospora* sp. fruit on deadwood and are successful in soils of young forest stands or in cooperation with seedlings [38,110] raises the question of whether these fungi could partly or temporarily use other C sources than plant photosynthates, which may be a possible explanation for their successful survival in gaps [47]. However, this ability has not been proven yet, and the saprobic lifestyle of ectomycorrhizal fungi is still a matter of debate [111,112]. Therefore, their survival on living tree roots that spread into soil in gaps from the surrounding vegetation seems to be more reasonable.

Our findings provide support to our hypothesis (2) that fungi will be tighter related with the time since disturbance, while bacterial assemblages were crucially shaped by soil pH and relative N availability. However, to interpret our data correctly, the bacterial community composition was also to a large extent dependent on the time since disturbance as the soil pH and C:N ratio were modified by the disturbance, and they were gradually adjusted by ecosystem regeneration and successional progression.

### 4.5. The Effect of Disturbance within the Soil Profile

In agreement with hypothesis (3), the effects related to disturbance and the following progression along the forest life cycle were more pronounced in the surface organic than deeper mineral soil horizons. In organic horizons, the type of vegetation cover was closely connected to the soil microbiome; thus both bacterial and fungal communities clearly differed between gaps and mature forest stands. This suggests that in a disturbance-driven development of the forest ecosystem, the key event is the transition from a tree-dominated forest stand to an open-canopy grass-covered gap and vice versa, with related changes in the quality of litter input. The gap opening allows the expansion of grasses, which increases soil nutrient ability and allows a shift from a specialized old-growth forest microbial community towards a more diverse microbiome with a higher proportion of copiotrophs and higher bacterial abundance. Then, canopy closure and replacement of grass- with spruce-derived low-quality litter reduces nutrient availability and triggers recovery of the original oligotrophic microbial community characteristic of old-growth forest ecosystems. This transition is the most rapid and pronounced in surface organic horizons.

In topsoil A horizons, containing microbially transformed organic matter bound to mineral particles, the disturbance-related effect was slightly delayed (the highest N availability at the 36-year site) and the transition between closing gaps and mature forests was less apparent. The microbial communities continually changed along the successional trajectory from the open gaps towards the sites being the longest time after the last disturbance (160 years).

The progression of the ecosystem from open gaps towards a mature forest with a closed canopy was also apparent in spodic horizons, but its manifestation was delayed by ca. 100 years compared to the topsoil. During that time, however, the E and Bhs horizons, which were absent in the 16–110-year sites, developed (see Figure 1). Their formation was surprisingly fast if we consider that the establishment of Albic Podzols with a typical sequence of horizons A-E-Bhs-Bs-BC-C usually takes millennia [113,114]. Such a rapid development points rather to a re-establishment of the horizons instead of their completely new formation, which could be an attached process accompanying the natural life cycle of the spruce forest (from the old-growth forest through open gaps and their succession back to the mature forest, driven by occasional disturbances). The Albic Podzol is a soil type characteristic of natural old-grown spruce forests (see also Figure 1). We speculate that a canopy opening and post-disturbance development of grassy vegetation (together with related changes in the microbiome and mediated soil processes) probably caused a deepening of A horizon, which covered and overgrew the original E and Bhs horizons. It resulted in the formation of the soil type classified as Entic Podzol. Subsequent gradual canopy closing led to a retreat of grasses, immobilization of available nutrients in growing woody vegetation, a change in the litter character and soil acidification, conditions supporting the rapid re-establishment of Albic Podzols. There are a few other reports on regressive Podzol development in the literature. Šamonil et al. [17] described the development of Albic Podzols from Entic Podzols, occurring in the order of hundreds of years instead of millennia, after soil mixing by tree uprooting. Regressive Podzol development due to the expansion of the grassland community without mechanical soil disturbance was described after the anthropogenic impact by Barret and Schaetzl [16]. However, as far as we know, such Podzol regression associated with potential soil rejuvenation has never been described as a component of a natural cycle of the terrestrial ecosystem. To increase our understanding of the relationship between forest dynamics and soil genesis, the transition from a mature closed forest to gaps, the event in forest dynamics that is crucial for the possible soil rejuvenation, requires further research.

## 5. Conclusions

The stand-replacing disturbance and subsequent development was successfully reflected in soil microbial communities. We discovered that during the forest life cycle, two key events in terms of soil microbial community succession occur. First, the enhanced

N availability and a moderate increase in pH due to the gap opening and expansion of the grassy understorey restructured the original old-growth forest microbiome. Second, the re-establishment of the closed-canopy forest enabled the recovery of the mature forest microbial community.

The initial phases of forest regeneration had higher bacterial abundance and diversity. But contrary to our expectations, there was no effect of the disturbance on the relative abundance of ectomycorrhizal fungi in the open forest gaps, which probably survived on the roots of the surrounding trees. Once the closed-canopy forest re-established and spruces became the main source of incoming organic material, the soils became gradually depleted in N, the pH decreased and the microbial community composition shifted back towards the dominance of specialized oligotrophs. The shifts in the microbiome structure were the most evident in the surface organic horizons, and the disturbance effect was weaker and delayed in the deeper soil horizons.

The forest life cycle was further likely accompanied with rejuvenation of the soils, based on the development of Entic Podzols in the gaps and relatively fast re-establishment of typical Albic Podzols in old-growth forests of the age of over 100 years.

**Supplementary Materials:** The following are available online at https://www.mdpi.com/article/10.3390/f12040410/s1. Figure S1: Canonical correlation analysis (CCA) of fungal OTUs based on variables that explained most of the variability in the fungal community composition in (A) O horizons, (B) A horizons and (C) B horizons. All fungal OTUs were included in the analysis and projected in the plot, but only the 25 most abundant OTUs across the respective horizon in blue are labelled; the size of blue points represents the relative abundance in %. Figure S2: Canonical correlation analysis (CCA) of bacterial OTUs based on variables that explained most of the variability in the bacterial community composition in (A) O horizons, (B) A horizons and (C) B horizons. All bacterial OTUs were included in the analysis, but only the 100 most abundant OTUs across the respective horizon are shown in the plot; the points are coloured according to the taxonomic classification, and the size of points represents the relative abundance in %. Figure S3: Relationship between DOC:DN and the relative abundance of microbes potentially capable of (A) nitrification, (B) N fixation and (C) nitrate respiration in the respective horizons. Pearson correlation coefficients and corresponding *p*-values indicated. Table S1: Soil chemical parameters that were available for selection when developing db-RDA models. Values represent the mean for the respective sites and horizons. Table S2: Site differences in the respective horizons for the most abundant fungal genera/orders (with a prevalent lifestyle) and all ectomycorrhizal fungi summed. Values represent the mean of relative abundance (in %) with standard deviation in parentheses. Small letters denote differences significant at *p* < 0.05. Table S3: Effect of site and O horizon type (PERMANOVA) on the fungal and prokaryotic community composition in respective horizons, model permutation *p*-values and explained variability. Table S4. Results of fungal community pairwise PERMANOVA comparisons: site differences for the respective horizons and pairwise comparison of O horizon types. Table S5: Site differences in the respective horizons for the most abundant bacterial classes. Values represent the mean of relative abundance (in %) with standard deviation in parentheses. Small letters denote differences significant at *p* < 0.05. Table S6. Results of fungal community pairwise PERMANOVA comparisons: site differences for the respective horizons and pairwise comparison of O horizon types. Table S7. Distance-based redundancy analysis (db-RDA) models best explaining the variability in the fungal and bacterial community composition in respective horizons, model permutation *p*-values, total model-explained variability, explanatory variables and proportion of total variability explained by particular variables.

**Author Contributions:** P.Š. and H.Š. designed the study. P.Š. and M.V. collected the samples. M.V. and H.Š. funded the analyses. M.C. and J.B. processed the microbiome data. M.C. performed statistical analyses. M.C. and P.Š. wrote the manuscript. E.K., K.T., J.B., M.V. and H.Š. critically reviewed and edited the manuscript. All authors have read and agreed to the published version of the manuscript.

**Funding:** The study was financed by the Czech Science Foundation (project nos. 15-14840S (field sampling and soil analyses), 19-16605S (microbiome data analysis) and 19-09427S (contribution of P.Š.)).

**Data Availability Statement:** Sequencing data are available in a publicly accessible repository. The other data are available upon request.

**Acknowledgments:** The authors thank Standa Kašík for help in data collection. The authors are also grateful to the Calimani National Park administration for enabling the implementation of the project.

**Conflicts of Interest:** The authors declare no conflict of interest.

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
