# Peer review of "Soil Microbiome Composition along the Natural Norway Spruce Forest Life Cycle"

_forests, doi:10.3390/f12040410_

Round 1
Reviewer 1 Report
This manuscript examined the variation of soil properties and soil microbiome composition along the chronosequence of forest life cycle. It is an interesting research, but the writing style need further improve. There are several issues that would have to be addressed.
- There are too many the introduction of the stand-replacing disturbances in the section of introduction, which should convince on the novelty of the work and this can only be done by (1) acknowledging the existing literature about forest life cycle and its effect on soil properties, soil microbiome composition; what has be done so far on the subject by referring to existing research studies with quantitative information, and reporting ones; referring to methods and ideas associated with other researchers; (2) discussing the existing finding and identifying research gap(s); critically discuss these existing works to summarize research advances and gaps in knowledge of the subject (how changes in soil microbiome composition are associated to changes on soil environment or forest life cycle? Is there a link? (3) clearly state the research objectives, which should be in accordance with the identified gaps. Finally, the discussion section should compare the obtained results with the existing literature.
Tips for scientific writing of the introduction
- Presenting the background of the subject;
- Indicating the importance of the research on the subject;
- Acknowledging what has be done so far on the subject by referring to existing research studies and reporting ones; referring to methods and ideas associated with other researchers;
- Pointing to a gap in knowledge of the subject;
- Selecting research objectives
- Explaining the organization of the research.
2. In the section of Materials and Methods
I think a sound chronosequence should include 16, 50, 70, 100, 130, 160 years after the disturbance, I know it is hard to find the dream treatment, but the disturbance treatment in this research was at the both two ends of disturbance.
When and where were the soils sampled?
Please move the sentence in lines 217-220 to the note section of table 1.
- In the section of results
The description should be concise, I think the tables 4 and 5 is too big, please list the significant data in table.
In lines 369-381: the description about the soil microbiome composition is too much, the authors should put more effort on the variations of soil microbiome composition among site or forest life cycle treatment.
There were no numbers for the axis in figures 2 and 3.
Author Response
Reviewer 1
Open Review
(x) I would not like to sign my review report
( ) I would like to sign my review report
English language and style
( ) Extensive editing of English language and style required
(x) Moderate English changes required
( ) English language and style are fine/minor spell check required
( ) I don't feel qualified to judge about the English language and style
Yes Can be improved Must be improved Not applicable
Does the introduction provide sufficient background and include all relevant references?
( ) ( ) (x) ( )
Is the research design appropriate?
( ) (x) ( ) ( )
Are the methods adequately described?
(x) ( ) ( ) ( )
Are the results clearly presented?
( ) (x) ( ) ( )
Are the conclusions supported by the results?
( ) (x) ( ) ( )
Comments and Suggestions for Authors
This manuscript examined the variation of soil properties and soil microbiome composition along the chronosequence of forest life cycle. It is an interesting research, but the writing style need further improve. Major revisions and a second review are necessary before this manuscript is suitable for publication. There are several issues that would have to be addressed.
There are too many the introduction of the stand-replacing disturbances in the section of introduction, which should convince on the novelty of the work and this can only be done by (1) acknowledging the existing literature about forest life cycle and its effect on soil properties, soil microbiome composition; what has be done so far on the subject by referring to existing research studies with quantitative information, and reporting ones; referring to methods and ideas associated with other researchers; (2) discussing the existing finding and identifying research gap(s); critically discuss these existing works to summarize research advances and gaps in knowledge of the subject (how changes in soil microbiome composition are associated to changes on soil environment or forest life cycle? Is there a link? (3) clearly state the research objectives, which should be in accordance with the identified gaps. Finally, the discussion section should compare the obtained results with the existing literature.
Tips for scientific writing of the introduction
Presenting the background of the subject;
Indicating the importance of the research on the subject;
Acknowledging what has be done so far on the subject by referring to existing research studies and reporting ones; referring to methods and ideas associated with other researchers;
Pointing to a gap in knowledge of the subject;
Selecting research objectives
Explaining the organization of the research.
In response to concerns raised by Reviewer 1, we thoroughly revised the introduction and believe that now its structure and content better fit above provided general structure recommendations. We reduced the opening part regarding general information about forest disturbances and put more emphasis on the context of complete forest life cycle. Then we continue with changes in vegetation due to disturbance and ecosystem regeneration therefrom and their effect on soil properties. We describe links between changing soil properties and soil microbiome composition and finally we provide an overview of existing research in this field and name the existing gaps in respect to soil microbiome development along natural forest life cycle.
- In the section of Materials and Methods
I think a sound chronosequence should include 16, 50, 70, 100, 130, 160 years after the disturbance, I know it is hard to find the dream treatment, but the disturbance treatment in this research was at the both two ends of disturbance.
We agree that a more intensive and more evenly distributed coverage along chronosequence as well as multiple independent sites for the respective stand ages (or phases in forest succession) would be a more ideal design for conducting such study. Unfortunately, real possibilities are usually far from ideal. Primeval forest is not a laboratory where we can design our experiment in all aspects. Here, we were limited by findings of previous studies regarding forest disturbance regime (Svoboda et al. 2014) and soil variability and diversity (Valtera et al. 2013, 2015). All these studies were comparatively large in terms of analysed data and represented solid base for our research. We believe that we did our best to select the most representative and extensive set of available sites in the process that was described in the M&M section. We acknowledge that our chronosequence misses sites that would fall into interval of 40-100 years of post-disturbance development, but we believe that despite this drawback, our study provides a valuable insight into this understudied field of forest soil microbiome ecology.
Svoboda et al. (2014). Landscape-level variability in historical disturbance in primary Picea abies mountain forests of the Eastern Carpathians, Romania. Journal of Vegetation Science, 25(2), 386–401. https://doi.org/10.1111/jvs.12109
Valtera, M., Šamonil, P., & Boublík, K. (2013). Soil variability in naturally disturbed Norway spruce forests in the Carpathians: Bridging spatial scales. Forest Ecology and Management, 310, 134–146. https://doi.org/10.1016/j.foreco.2013.08.004
Valtera et al. (2015). Effects of topography and forest stand dynamics on soil morphology in three natural Picea abies mountain forests. Plant and Soil, 392(1–2), 57–69. https://doi.org/10.1007/s11104-015-2442-4
When and where were the soils sampled?
We completed the information in the manuscript.
Lines 172-173:
“The research took place in volcanic Calimani Mts. in the Inner Eastern Carpathians, northern Romania (47°6.115′ N, 25°7.129′ E) in July 2015.”
Please move the sentence in lines 217-220 to the note section of table 1.
In our opinion, this information is particularly important for readers to better understand description and discussion of soil microbiome composition changes in O horizons along the chronosequence and therefore we decided to keep this sentence in the main body.
In the section of results
The description should be concise, I think the tables 4 and 5 is too big, please list the significant data in table.
We agree that the extensive tables in the article main body might by overwhelming. We reduced the tables presented within the main body and provide their original full versions in the supplement. Now, Table 4 lists only ectomycorrhizal genera as this group is the most discussed in this manuscript. Table 5 newly presents only bacterial classes that show significant site differences in any of studied horizons.
In lines 369-381: the description about the soil microbiome composition is too much, the authors should put more effort on the variations of soil microbiome composition among site or forest life cycle treatment.
We considered this suggestion carefully, but we have the opinion that before the description of results can focus on the differences in microbiome composition along the chronosequence, readers need to be provided with a general overview of the communities’ composition (now it is even more important to give such description in the text when Tables 4 and 5 were reduced based on the previous suggestion). Reviewer 1 further suggests putting more effort on explaining variations in soil microbiome composition among sites. To keep this study as straightforward as possible, we tried to avoid giving an overwhelming list of all differences between sites that might be found in the data, but we rather aimed to discuss only the most significant and relevant results. We believe that this approach makes the best compromise, which results in a comprehensive, but also straightforward manuscript.
There were no numbers for the axis in figures 2 and 3.
We added axis tick mark labels in both Figure 2 and 3.
Submission Date
26 February 2021
Date of this review
15 Mar 2021 02:49:25

Reviewer 2 Report
Dear authors,
I have read the article with interest and the following questions come to mind. Has the spruce stand been growing on forest land for many generations or was it established on former farmland? Can the analysis of the bacteriome analysis help to answer this question? Were species characteristic of cultivated plants or grazing cattle found?
Concerning fungi, the main pathogens present in the soil of spruce stands and destroying their root systems are Heterobasidion parviporum and fungi of the genus Armillaria. Were they not found at all in the study or would this require different sampling techniques?
Was the spruce stand in the park completely protected or was it subject to partial forest management and periodic treatments (cleaning, thinning, felling) were carried out there?
Can we talk about soil rejuvenation after a stand of trees is exposed when grasses appear? Foresters then speak of soil/station degradation as a consequence of massive weed growth and the associated difficulties in making such forest areas productive (regeneration). In addition, the population of mice, which damage the planted seedlings, is increasing and their population needs to be reduced in order to restore the forest.
L49-50 – “The stand-replacing disturbances are characteristically caused by abiotic (e.g wind-49 storm) and biotic (typically a bark beetle outbreak) factors in old-growth 50 spruce-dominated mountain forests.” – for this statement (and below) you can cite: Nowakowska, J. A., Hsiang, T., Patynek, P., Stereńczak, K., Olejarski, I., & Oszako, T. (2020). Health assessment and genetic structure of monumental Norway spruce trees during a bark beetle (Ips typographus L.) outbreak in the Białowieża Forest District, Poland. Forests, 11(6), 647.
L169 – Picea abies – put in Italic
L171 – m2 change to m2
L260 – add space after [64], before and
L445 – is CCA explained somewhere? As wel as DOC:DN
L487 what do you mean both sensu?
Author Response
Reviewer 2
Open Review
(x) I would not like to sign my review report
( ) I would like to sign my review report
English language and style
( ) Extensive editing of English language and style required
( ) Moderate English changes required
(x) English language and style are fine/minor spell check required
( ) I don't feel qualified to judge about the English language and style
Yes Can be improved Must be improved Not applicable
Does the introduction provide sufficient background and include all relevant references?
(x) ( ) ( ) ( )
Is the research design appropriate?
(x) ( ) ( ) ( )
Are the methods adequately described?
(x) ( ) ( ) ( )
Are the results clearly presented?
(x) ( ) ( ) ( )
Are the conclusions supported by the results?
(x) ( ) ( ) ( )
Comments and Suggestions for Authors
Dear authors,
I have read the article with interest and the following questions come to mind. Has the spruce stand been growing on forest land for many generations or was it established on former farmland? Can the analysis of the bacteriome analysis help to answer this question? Were species characteristic of cultivated plants or grazing cattle found?
The current research builds on the previous research of the forest disturbance history, in which only stands in core zone of Calimani National Park (CNP) without any signs of direct human intervention were included and these forest stands were considered to be primeval (Svoboda et al. 2014). This is far from being the predominant character of the forest vegetation in the CNP. Exactly opposite. Most of the forests in the CNP contain clear signs of historical logging, cattle grazing, etc. At the sites of our research, however, we assume a very minimal historical direct human intervention. The results of the study of the disturbance past supported this view. For details see: Svoboda et al. (2014). Landscape-level variability in historical disturbance in primary Picea abies mountain forests of the Eastern Carpathians, Romania. Journal of Vegetation Science, DOI:10.1111/jvs.12109 (the most relevant part copy-pasted below):
“These [potential study] stands were (…) surveyed for indicators of naturalness (e.g. coarse woody debris in various stages of decay, pit-and-mound topography) and signs of human impact; stands with evidence of past logging and grazing were avoided, as were stands in close proximity to formerly grazed areas. Additionally, we searched all the available archival information regarding the history of land use in these areas. Historical data indicate the areas selected for study escaped logging in the 18th and 19th centuries and have been protected since this time.”
Concerning fungi, the main pathogens present in the soil of spruce stands and destroying their root systems are Heterobasidion parviporum and fungi of the genus Armillaria. Were they not found at all in the study or would this require different sampling techniques?
Some samples contained sequences of Heterobasidion sp., but only in negligible counts (<10 sequences per sample, <0.01% in relative abundance), while no Armillaria sp. sequences were detected. We primarily aimed to describe the microbial communities in free soil, not on/in the roots and we excluded any present visible roots when sampling the soil. That is very likely the culprit of negligible detection of this fungal guild in our dataset, as these fungi proliferate bulk soil to a considerably lesser extent compared to saprotrophic and mycorrhizal fungi. In case of studying presence and diversity of such species, the direct sampling of tree roots would be more effective compared to our approach.
Was the spruce stand in the park completely protected or was it subject to partial forest management and periodic treatments (cleaning, thinning, felling) were carried out there?
To our knowledge, the area that was subject of this study is the most preserved core zone in Calimani NP and no anthropogenic management has not been performed here for decades. Newly, we emphasise this in the opening part of the Discussion section.
Lines 514-517:
“The studied chronosequence consisted of six primeval Norway spruce forest sites located in the most preserved core area of the Calimani National Park, where the ecosystems naturally evolved since the last stand-replacing disturbance without any signs of direct human intervention [50].”
Can we talk about soil rejuvenation after a stand of trees is exposed when grasses appear? Foresters then speak of soil/station degradation as a consequence of massive weed growth and the associated difficulties in making such forest areas productive (regeneration). In addition, the population of mice, which damage the planted seedlings, is increasing and their population needs to be reduced in order to restore the forest.
Some misunderstanding probably arose here and two different things are mixed. The term soil rejuvenation concerns exclusively changes in soil properties (decreasing soil acidity or increasing amount of nutrients). So, it can be used also when grasses appear in forest if observed response in soil chemical properties support such evaluation. The expansion of grasses might be from a silvicultural point of view indeed regarded as “site/stand degradation”, as it might impede both natural and artificial regeneration. However, from the point of view of soil acidity and nutritional status (which also affect stand regeneration and productivity), it usually has a positive effect. The reversal of acidification and nutrient depletion by grass expansion might be viewed as a reciprocal force to soil genesis, its ageing (podzolization in spruce forests). Therefrom the term “rejuvenation”. We believe that we can talk about “soil rejuvenation” when grasses spread in an opened forest stand and the soil nutritional status is improved. Our data suggest that such process of rejuvenation might be present in forest soils as part of the forest life cycle, however we fully acknowledge that this potential process warrants further research to satisfactorily confirm or reject its hypothetical presence and nature.
L49-50 – “The stand-replacing disturbances are characteristically caused by abiotic (e.g wind-49 storm) and biotic (typically a bark beetle outbreak) factors in old-growth 50 spruce-dominated mountain forests.” – for this statement (and below) you can cite: Nowakowska, J. A., Hsiang, T., Patynek, P., Stereńczak, K., Olejarski, I., & Oszako, T. (2020). Health assessment and genetic structure of monumental Norway spruce trees during a bark beetle (Ips typographus L.) outbreak in the Białowieża Forest District, Poland. Forests, 11(6), 647.
Thank you for this suggestion, we added this citation.
Line 43.
L169 – Picea abies – put in Italic
Corrected.
Line 184.
L171 – m2 change to m2
Corrected.
Line 186.
L260 – add space after [64], before and
Corrected.
Line 276.
L445 – is CCA explained somewhere? As wel as DOC:DN
Both these abbreviations were explained in the preceding section of Materials and Methods. To make the text easier to follow, we newly explain these terms in their first occurrence in the Results section as well.
Lines 349-350:
“Soil dissolved organic C (DOC) to DN ratio (DOC:DN) increased…”
Line 450:
“The canonical correspondence analysis (CCA) plot suggests that…”
L487 what do you mean both sensu?
That was supposed to mean that both types of organic layers (the mature forest LFH mor layer and grass derived T horizon were described according to (i.e. sensu) cited reference. We rephrased this to avoid confusions.
Line 524-527:
“The typical spruce forest mor layer consisting of decomposing needles in thick L, F, and especially H horizons was in these gaps replaced by comparatively thin T horizon originating from tissues of living graminoids and their residues [described according to 57].”
Submission Date
26 February 2021
Date of this review
12 Mar 2021 13:31:18

Round 2
Reviewer 1 Report
The authors provide a good revision, there is no specific comment.